# Inhibitory Effect and Mechanism of Action of Quercetin and Quercetin Diels-Alder *anti*-Dimer on Erastin-Induced Ferroptosis in Bone Marrow-Derived Mesenchymal Stem Cells

**DOI:** 10.3390/antiox9030205

**Published:** 2020-03-02

**Authors:** Xican Li, Jingyuan Zeng, Yangping Liu, Minshi Liang, Qianru Liu, Zhen Li, Xiaojun Zhao, Dongfeng Chen

**Affiliations:** 1School of Chinese Herbal Medicine, Guangzhou University of Chinese Medicine, Waihuan East Road No. 232, Guangzhou Higher Education Mega Center, Guangzhou 510006, China; zengjingyuan710@163.com (J.Z.); 20181105440@stu.gzucm.edu.cn (M.L.); liuqianru2333@163.com (Q.L.); zxj@gzucm.edu.cn (X.Z.); 2Innovative Research & Development Laboratory of TCM, Guangzhou University of Chinese Medicine, Waihuan East Road No. 232, Guangzhou Higher Education Mega Center, Guangzhou 510006, China; 3The Fourth Clinical Medical College, Guangzhou University of Chinese Medicine, Waihuan East Road No. 232, Guangzhou Higher Education Mega Center, Guangzhou 510006, China; 4Department of Anatomy, The Research center of Basic Integrative Medicine, Guangzhou University of Chinese Medicine, Guangzhou 510006, China; 20182104114@stu.gzucm.edu.cn

**Keywords:** Diels-Alder dimer, quercetin, antioxidant, QDAD, anti-ferroptosis, erastin

## Abstract

In this study, the anti-ferroptosis effects of catecholic flavonol quercetin and its metabolite quercetin Diels-Alder *anti*-dimer (QDAD) were studied using an erastin-treated bone marrow-derived mesenchymal stem cell (bmMSCs) model. Quercetin exhibited higher anti-ferroptosis levels than QDAD, as indicated by 4,4-difluoro-5-(4-phenyl-1,3-butadienyl)-4-bora-3a,4a-diaza-s-indacene-3-undecanoic acid (C11-BODIPY), 2′,7′-dichlorodihydrofluoroscein diacetate (H2DCFDA), lactate dehydrogenase (LDH) release, cell counting kit-8 (CCK-8), and flow cytometric assays. To understand the possible pathways involved, the reaction product of quercetin with the 1,1-diphenyl-2-picrylhydrazyl radical (DPPH^●^) was measured using ultra-performance liquid-chromatography coupled with electrospray-ionization quadrupole time-of-flight tandem mass spectrometry (UHPLC-ESI-Q-TOF-MS). Quercetin was found to produce the same clusters of molecular ion peaks and fragments as standard QDAD. Furthermore, the antioxidant effects of quercetin and QDAD were compared by determining their 2-phenyl-4,4,5,5-tetramethylimidazoline-1-oxyl-3-oxide radical-scavenging, Cu^2+^-reducing, Fe^3+^-reducing, lipid peroxidation-scavenging, and DPPH^●^-scavenging activities. Quercetin consistently showed lower IC_50_ values than QDAD. These findings indicate that quercetin and QDAD can protect bmMSCs from erastin-induced ferroptosis, possibly through the antioxidant pathway. The antioxidant pathway can convert quercetin into QDAD—an inferior ferroptosis-inhibitor and antioxidant. The weakening has highlighted a rule for predicting the relative anti-ferroptosis and antioxidant effects of catecholic flavonols and their Diels-Alder dimer metabolites.

## 1. Introduction

Ferroptosis is a regulated cell death process that depends on iron [1,2]. It is directly mediated by the accumulation of lipid peroxidation (LPO) and reactive oxygen species (ROS) [3,4,5]. On the one hand, the positive modulation of ferroptosis can be used to procedurally kill cancer cells, and thus offers a potential approach for cancer therapy [1,2]. On the other hand, the negative modulation of ferroptosis can inhibit cellular death to improve the feasibility of transplantation application to treat ferroptosis-related diseases, e.g., Parkinson’s disease [6,7,8]. As a result, discovering novel ferroptosis inhibitors has become a hot pursuit in cell biology and chemical biology [8,9].

Recently, some synthetic radical scavengers have been reported to inhibit ferroptosis, such as α-tocopherol [10] and hydroxylated chalcones [11]. In addition, some iron chelator (e.g., deferiprone and deferoxamine) and reducing (e.g., *β*-mercaptoethanol) reagents could also act as ferroptosis inhibitors [10]. The radical-scavenging, iron-chelating, and reducing properties are the three characteristics of natural antioxidants. This provides a clue that effective natural antioxidants may be potential ferroptosis inhibitors.

Therefore, the present study selected quercetin as a model for anti-ferroptosis exploration. Quercetin has been demonstrated to be an antioxidant flavonol and has played a beneficial (or cytoprotective) role in various lines in recent years (Appendix A) [12,13,14,15,16,17,18,19,20,21]. Structurally, quercetin bears a 2,3-C=C bond, 3-OH, and 3′,4′-di-OHs (i.e., 3′,4′-catecholic moiety), thus, it belongs to typical catecholic flavonol (Figure 1A). This catecholic flavonol can be enzymatically transformed into a quercetin dimer (i.e., quercetin Diels-Alder *anti*-dimer, QDAD, Figure 1B) during plant metabolism [22,23,24,25,26]. Because of this, quercetin and its dimer metabolite QDAD usually co-exist in the same plant [27]. The present study thus tried to comparatively evaluate anti-ferroptosis effects of quercetin and QDAD.

The ferroptosis model was created by a small molecule erastin in the study. This is because erastin can block the biosynthesis of glutathione, which is an endogenous antioxidant in cells [5,8] that results in an impairment of the redox balance and an accumulation of LPO in the cells. Cellular iron (specially Fe^2+^) catalyzes LPO into the LOO^●^ radical and other ROS, causing oxidative damage to cells, and leading to cell death [3,4,5]. Erastin has recently been used to create a ferroptosis model in SH-SY5Y neuroblastoma cells and tumor cells [11,28,29]. The present study attempted to use bone marrow-derived mesenchymal stem cells (bmMSCs) as model cells. The bmMSCs are known as important seed cells for transplantation engineering to treat degenerative diseases, such as Parkinson’s disease and geriatric diseases [30,31]. The introduction of bmMSCs apparently strengthens the clinical application of the study.

To understand the possible mechanisms, quercetin was mixed with the DPPH^●^ radical—a mimic of ROS or the LOO^●^ radical [32]; the product mixture was then chemically analyzed using cut-edging ultra-performance liquid chromatography coupled with electrospray ionization quadrupole time-of-flight tandem mass spectrometry (UHPLC-ESI-Q-TOF-MS) technology. UHPLC-ESI-Q-TOF-MS analysis can bring about key evidence for whether quercetin has been oxidized into QDAD when interacting with free radicals.

The present study tries to combine cellular and chemical approaches to explore the inhibitory effect and mechanisms of action of quercetin and QDAD on erastin-induced ferroptosis in bmMSCs. It is expected to offer novel and reliable information for cell biology, chemical biology, metabolomics, stem cell transplantation, and antioxidant chemistry. The information will also benefit the understanding of catecholic flavonol—a natural product family containing at least 75 members [33,34] (some of them are listed in Appendix A).

## 2. Materials and Methods

### 2.1. Animals, Cellular Kits, and Chemicals

Four-week-old Sprague-Dawley (SD) rats were obtained from the Animal Center of Guangzhou University of Chinese Medicine. The animal experiment was performed in strict accordance with the regulations of the Institutional Animal Ethics Committee in Guangzhou University of Chinese (Approval number 20180034).

Fetal bovine serum (FBS), rat bone mesenchymal stem cell Basal medium, and trypsin were purchased from Gibco (Grand Island, NY, USA). 4,4-Difluoro-5-(4-phenyl-1,3-butadienyl)-4-bora-3a,4a-diaza-s-indacene-3-undecanoic acid (C11-BODIPY) was obtained from Invitrogen (Carlsbad, CA, USA). Percoll was obtained from GE Healthcare Life Sciences (Pittsburgh, PA, USA). Trypsin was from Promega Co. (Madison, WI, USA). Erastin was purchased from Bertin Bioreagent (Montigny-le-Bretonneux, France). Ferrostatin-1 was purchased from Selleck Chemicals (Houston, TX, USA); H2DCFDA was from Invitrogen Co. (Waltham, MA, USA). LDH release reagent was obtained from Promega Corporation (Madison, WI, USA). 4′,6-diamidino-2-phenylindole (DAPI) was from SouthernBiotech Co. (Birmingham, AL, USA). BBcell Probe^TM^ was from Best Bio Co. (Shanghai, China). Stem cell Basal medium RASMX-90041 was obtained from Cyagen Biosciences Co. (Guangzhou, China).

Quercetin (C_15_H_10_O_7_, CAS number 117-39-5, MW 302.2, purity 98%, Appendix A) was obtained from Biopurify Phytochemicals, Ltd. (Chengdu, China). QDAD (C_30_H_18_O_14_, CAS number 167276-19-9, MW 602.4, purity 98%, Appendix A) was obtained from BioBioPha, Ltd. (Kunming, China). (±)-6-Hydroxyl-2,5,7,8-tetramethylchromane-2-carboxylic acid (Trolox), 1,1-diphenyl-2-picrylhydrazyl radical (DPPH^•^), 2,4,6-tripyridyltriazine (TPTZ), linoleic acid, and 2,9-dimethyl-1,10-phenanthroline (neocuproine) were obtained from Sigma-Aldrich (Shanghai, China). 2-Phenyl-4,4,5,5-tetramethylimidazoline-1-oxyl-3-oxide radical (PTIO^•^) was obtained from TCI Chemical Co. (Shanghai, China). (−)-Ascorbic acid was obtained from J&K Scientific (Beijing, China). Hydrogen peroxide (H_2_O_2_, AR), ferrous chloride (FeCl_2_∙4H_2_O, AR), and dimethyl sulfoxide (DMSO, AR) were obtained from Guangzhou Chemical Reagent Factory (Guangzhou, China). Water, methanol, and acetonitrile of HPLC grade, and other reagents of analytical grade, were purchased from Guangdong Guanghua Chemical Plants Co., LTD (Shantou, China).

### 2.2. Characterization of Mitochondrial ROS in Ferroptotic bmMSCs

Bone marrow was obtained from the femur and tibia of rats according to a previously published method [35]. In brief, the marrow was diluted with rat bone mesenchymal stem cell basal medium containing 10% FBS and then subjected to gradient centrifugation at 900× *g* for 30 min on 1.073 g/mL Percoll to obtain bmMSCs. The bmMSCs were detached with 0.25% trypsin and cultured in flasks at 1 × 10^4^ cells/cm^2^ to characterize the mitochondrial ROS concentration [36,37,38].

In the experiment, bmMSCs were seeded at 1 × 10^6^ cells/well into 12-well plates. After 24 h contact, bmMSCs were divided into four groups, i.e., control, model, positive control, and sample groups. In the control group, bmMSCs were incubated for 12 h in stem cell basal medium. In the model, positive control, and sample groups, bmMSCs were incubated in the presence of erastin (10 μM). Following a 12 h incubation period, the mixture of erastin and medium was removed. The model group was incubated for 12 h in stem cell basal medium, while the sample group was incubated for 12 h in the same medium, with the addition of 0.03 μM quercetin (or QDAD), and the positive control was incubated for 12 h in the same medium, with 1.0 μM ferrostatin-1.

Furthermore, the bmMSCs were incubated for 10 min with 4′,6-diamidino-2-phenylindole (DAPI) staining, followed by a 30 min incubation period, to determine mitochondrial ROS with the H2DCFDA probe (5 μM) and a mitochondrial fluorescent probe [39].

### 2.3. Flow Cytometry with the C11-BODIPY Probe to Assess LPO Accumulation

The bmMSCs obtained according to Section 2.2 were evaluated for LPO accumulation via flow cytometry and the C11-BODIPY probe [38,40]. The cells were then seeded at 1 × 10^6^ cells/well into 12-well plates. After 24 h contact, bmMSCs were divided into four groups, i.e., control, erastin, erastin + ferrostatin, and sample groups, according to a previous study [41]. In the control group, bmMSCs were incubated for 12 h in stem cell basal medium. In the erastin and sample groups, bmMSCs were incubated in the presence of erastin (10 μM). In the erastin + ferrostatin group, bmMSCs were incubated in the presence of erastin (10 μM) and ferrostatin-1 (1μM). After incubation for 12 h, the mixture of erastin and medium was removed. The erastin and erastin + ferrostatin groups were incubated for 12 h in stem cell basal medium, while the sample groups were incubated for 12 h in stem cell basal medium at different sample concentrations. Following incubation, the cells were rinsed twice with PBS. Adhered cells were incubated with C11-BODIPY (2.5 µM) staining solution in PBS in the dark for 30 min at 37 °C. Cells were then harvested with 0.05% trypsin solution, suspended in fresh medium, and immediately analyzed with a flow cytometer. Each sample test was repeated in three independent wells.

### 2.4. Flow Cytometry Using the H2DCFDA Probe to Determine the Total Intracellular ROS Concentration

The bmMSCs in the above-mentioned four groups (Section 2.3) were used to determine the total intracellular ROS concentration using flow cytometry and the H2DCFDA probe [41,42]. Briefly, bmMSCs were rinsed twice with PBS and then incubated with H2DCFDA (5 µM) staining solution in PBS in the dark for 30 min at 37 °C. Cells were harvested with 0.05% trypsin solution, suspended in fresh medium, and immediately analyzed with a flow cytometer. Each sample test was repeated in three independent wells.

### 2.5. Lactate Dehydrogenase (LDH) and CCK-8 Assays

The LDH release assay is based on a published method [43]. In brief, bmMSCs (Section 2.2) were seeded at 1 × 10^4^ cells/well into 96-well plates. After 24 h of contact, bmMSCs were divided into control, erastin, erastin + ferrostatin, and sample groups, according to a previous study [41]. The bmMSCs in the sample group were incubated for 12 h in stem cell basal medium at different chemical concentrations. Then, the mixture of chemicals and medium was removed. In the control group, bmMSCs were incubated for 12 h in stem cell basal medium. In the erastin and sample groups, bmMSCs were incubated in the presence of erastin (20 μM). In the erastin + ferrostatin groups, bmMSCs were incubated in the presence of erastin (20 μM) and ferrostatin (1 μM). Following 11 h incubation, the LDH release reagent was added to the maximum enzyme activity to each group for 1 h. After 1 h, the plate was centrifuged for 5 min with a porous plate centrifuge at 400× *g*. A volume of 120 μL was removed from each well and added to the well of a new 96-well plate. The LDH detection solution (60 μL) was added into each well and the plate was incubated for an additional 30 min. Absorbance at 490 nm was determined on a Bio-Kinetics reader (PE-1420; Bio-Kinetics Corporation, Sioux Center, IA, USA).

For the CCK-8 assay, the cells incubated for 11 h were treated with 90 μL RASMX-90011 and 10 μL CCK-8 solution for 2 h. The absorbance at 450 nm was determined on a Bio-Kinetics reader.

### 2.6. Colorimetric Measurement of PTIO^●^-Scavenging

The PTIO^●^-scavenging colorimetric measurement was conducted in accordance with our method [44]. Briefly, the test sample solution (*x* = 0–10 μL, 1 mg/mL) was added to (20 − *x*) μL of methanol, followed by 80 μL of an aqueous PTIO^●^ solution. The mixture was kept at 40 °C for 2 h, and the absorbance was then measured at 560 nm using a microplate reader (Multiskan FC, Thermo Scientific, Shanghai, China). The PTIO^●^ inhibition percentage was calculated as Formula (1),
(1)$$Inhibition\;\%=\frac{A_{0}-A}{A_{0}}\times 100\%$$
where *A*_0_ is the absorbance of the control without the sample, and *A* is the absorbance of the reaction mixture with the sample.

### 2.7. Colorimetric Measurement of DPPH^●^-Scavenging

The DPPH^●^ radical scavenging activity was determined based on a previously reported method [45]. Briefly, 80 μL of DPPH^●^ solution (0.1 mM) was added to a 96-well plate and mixed with samples of different concentrations (0–50 μg/mL). The total volume was then adjusted to 100 μL with methanol. The mixture was kept in the dark at 25 °C for 30 min, and the absorbance was then measured at 519 nm on a microplate reader (Multiskan FC, Thermo Scientific, Shanghai, China). The DPPH^•^ inhibition percentages of the samples were calculated using Formula (1).

### 2.8. Colorimetric Measurement of the Cupric Ion (Cu^2+^)-Reducing Antioxidant Capacity

The cupric ion-reducing antioxidant capacity (Cu^2+^-reducing) colorimetric measurement was conducted based on a previous method [46], with small modifications, as described in the report by Xie [47]. Twelve microliters CuSO_4_ solution (0.01 M) and 12 μL ethanolic neocuproine solution (7.5 mM) were added to a 96-well plate and mixed with different concentrations of samples (4–20 μg/mL). The total volume was then adjusted to 100 μL with a CH_3_COONH_4_ buffer solution (0.1 M) and mixed again to homogenize the solution. The mixture was maintained at room temperature for 30 min, and the absorbance was measured at 450 nm on a microplate reader (Multiskan FC, Thermo Scientific, Shanghai, China). The relative reducing power of the sample was calculated as follows:(2)$$Relative\;reducing\;power\%=\frac{A-A_{min}}{A_{max}-A_{min}}\times 100\%$$ where *A_max_* is the maximum absorbance, *A_min_* is the minimum absorbance, and *A* is the absorbance of the sample.

### 2.9. Colorimetric Measurement of the Ferric (Fe^3+^)-Reducing Ability

The ferric ion-reducing ability colorimetric measurement was conducted based on a previous method, with minor modifications [48,49]. In the present study, 20 mM FeCl_3_ was mixed with 10 mM TPTZ and 0.25 M acetate buffer (pH 3.6) at 1:1:10 to prepare the work solution. The sample solution (*x* = 1–9 μL, 0.1 mg/mL) was added to (20 − *x*) μL of 95% ethanol, followed by 80 μL of work solution. After incubation at ambient temperatures for 30 min, the absorbance was measured at 595 nm, using distilled water as the blank. The relative reducing power of the sample was calculated using Formula (2).

### 2.10. Colorimetric Measurement of Lipid Peroxidation-Scavenging (Linoleic Acid Emulsion Method)

Linoleic acid emulsion was selected as the specific substrates of ferroptosis. The linoleic acid emulsion was prepared based on a previous study [50]. Briefly, 1500 μL linoleic acid emulsion was mixed with 150 μL sample methanol solution and 350 μL 30% ethanol (*v*/*v*). The reaction mixture (total 2000 μL) was incubated at room temperature for 72 h. Then, 150 μL of the mixture was added to 3650 μL 75% ethanol (*v*/*v*), 100 μL NH_4_SCN (30%, *w*/*w*), and 100 μL FeCl_2_ (0.02 M in 3.6% HCl). Absorption at 500 nm was measured with a Unico 2100 spectrophotometer (Unico Co., Shanghai, China). The inhibition percentage was calculated by the following equation:(3)$$\mathrm{Inhibition}\%=\frac{A_{0}-A}{A_{0}}\times 100\%$$
where A_0_ is the absorbance of the control without sample, and A is the absorbance of the reaction mixture with sample.

### 2.11. UHPLC-ESI-Q-TOF-MS Measurement

The reaction of DPPH^●^ with quercetin was carried out under the conditions described previously [51]. In brief, methanol solution of quercetin was mixed with methanol DPPH^●^ solution at a molar ratio of 1:2, respectively, and the resulting mixture was incubated for 24 h at room temperature. Subsequently, the product was passed through a 0.22 μm filter for UHPLC-ESI-Q-TOF-MS measurement.

The UHPLC-ESI-Q-TOF-MS measurement was conducted as mentioned previously [52]. The UHPLC-ESI-Q-TOF-MS apparatus was equipped with a Phenomenex Luna C_18_ column (2.1 mm i.d. × 100 mm, 1.6 μm, Phenomenex Inc., Torrance, CA, USA). The mobile phase was employed for elution of the system and consisted of a mixture of methanol (phase A) and 0.1% formic acid water (phase B). The column was eluted at a flow rate of 0.2 mL/min, with the following gradient elution program: 0–2 min, maintain 30% B; 2–10 min, 30%–0% B; 10–12 min, 0%–30% B. The sample injection volume was 3 μL. The Q-TOF-MS analysis was performed on a Triple TOF 5600*^+^* mass spectrometer (AB SCIEX, Framingham, MA, USA) equipped with an ESI source, which was run in the negative ionization mode. The scan range was set at 100–2000 Da. The system was run with the following parameters: ion spray voltage, −4500 V; ion source heater temperature, 550 °C; curtain gas pressure (CUR, N_2_), 30 psi; nebulizing gas pressure (GS1, Air), 50 psi; Tis gas pressure (GS2, Air), 50 psi. The declustering potential (DP) was set at −100 V, whereas the collision energy (CE) was set at −45 V, with a collision energy spread (CES) of 15 V. The above experiments were repeated using QDAD, instead of quercetin.

### 2.12. Preferential Conformational Analysis by Computational Chemistry and Molecular Weight Calculation

The preferential conformation was analyzed based on force fields by computational chemistry. In brief, the energy minimization of quercetin and QDAD was respectively calculated through molecular mechanics II (MM2) using the Chem3D Pro14.0 program (PerkinElmer, Waltham, MA, USA) [53,54,55,56]. The preferential conformation has been expressed using the molecular models in Figure 1C,D.

Q-TOF-MS analysis is characterized by highly accurate *m/z* values, particularly molecular weights. The molecular weight calculation based on the formula is vital for a comparison with the *m/z* values from the Q-TOF-MS analysis. In the present study, the molecular weight calculations were conducted based on the accurate relative atomic masses. The relative atomic masses of C, H, O, and N were 12.0000, 1.007825, 15.994915, and 14.003074, respectively [57].

### 2.13. Statistical Analysis

Each experiment was performed in triplicate and the data were recorded as the mean ± SD (standard deviation). The dose–response curves were plotted using Origin 6.0 professional software (OriginLab, Northampton, MA, USA). The IC_50_ value was defined as the concentration required for 50% radical inhibition (or relative reducing power) [58]. It was calculated by linear regression analysis and is expressed as the mean ± SD (*n* = 3). The linear regression was analyzed using Origin 6.0. Significant differences between the mean IC_50_ values were determined using one-way ANOVA and a t-test. The analysis was performed using SPSS software 13.0 (SPSS Inc., Chicago, IL, USA) for Windows. *p* < 0.05 was considered statistically significant.

## 3. Results and Discussion

In this study, bmMSCs were treated with the imidazole ketone erastin. Erastin was originally identified from the high-throughput screening of anticancer drugs, and it has been widely used to create ferroptosis models [29,59]. The erastin-treated bmMSCs were stained with 4,4-difluoro-5-(4-phenyl-1,3-butadienyl)-4-bora-3a,4a-diaza-s-indacene-3-undecanoic acid (C11-BODIPY). This staining has been used to assess the LPO degree in ferroptosis [8,28,60,61,62]. The C11-BODIPY staining revealed that the erastin-treated group of cells showed the darkest green color (Figure 2B), indicating that erastin induced severe LPO accumulation. The green fluorescence confirmed the successful creation of an erastin-induced ferroptosis model.

The bmMSC control group displayed no green fluorescence (Figure 2A), suggesting no LPO accumulation. Similar to the control group, the positive control group (ferrostatin-1) [63,64] also showed negligible green fluorescence (Figure 2C). This result indicates that LPO accumulation is effectively inhibited by ferrostatin-1—a widely used ferroptosis-inhibitor—by targeting LPO [10,63,64].

The quercetin and QDAD groups also inhibited LPO accumulation, as determined by fluorescence analysis and flow cytometry (Figure 2D,E and Figure 3). The 2′,7′-dichlorodihydrofluorescein diacetate (H2DCFDA) probe, however, suggested that quercetin and QDAD might also prevent total ROS accumulation in cells (Figure 4). Prevention against LPO and total ROS accumulation protected bmMSCs from cytotoxicity. Therefore, using the lactate dehydrogenase (LDH) release assay, the quercetin and QDAD groups were found to decrease cellular death (Figure 5A) and increase cellular viability (Figure 5B). Based on these cellular assays and previous findings [9,65], it can be summarized that both quercetin and QDAD showed a ferroptosis-inhibitory effect; however, quercetin was more effective than QDAD. This can be partly supported by previous findings indicating that quercetin inhibits oxidative stress in neuronal [12,13,14] and other cells [15].

To explore whether they can interact with LPO substrates (e.g., LOO^●^), quercetin and QDAD were respectively investigated using the linoleic acid emulsion method [66]. It was observed that both quercetin and QDAD) could resist the oxidation of linoleic acid to release the accumulated LPO (especially the LOO^•^ radical, Appendix A).

The LOO^•^ radical and other ROS are usually unstable [67]; hence, the study used a stable DPPH^•^ radical as a mimic to explore the behaviors of quercetin (or QDAD) during the ferroptosis-inhibition process. As seen in Figure 6B, quercetin produced a chromatogram peak which could further yield a cluster of molecular ion peaks (i.e., *m/z* 602 and 601) by means of electrospray-ionization in the mass spectra (MS) field. These *m/z* values are basically double the molecular weight of quercetin, indicating the dimeric reaction of quercetin. Coincidentally, the fragments of dimeric quercetin were equivalent to those of the standard QDAD (Figure 6E–I). According to the fragments (Figure 6F,I), the MS mass spectra could be elucidated as Figure 7.

As illustrated in Figure 7, a *retro* Diels-Alder cleavage broke the molecular ion peak ([M] *m/z* 602) into two fragments: *m/z* 299 and *m/z* 301. Precise subtractions between the two fragments were calculated as *m/z* 2.0122 (301.0321–299.0199, Figure 6F) or *m/z* 2.0171 (301.0346–299.0175, Figure 6I). The value of *m/z* 2.0122 (or 2.0171) indicated the loss of two hydrogen atoms (2H). The relative atomic mass of the hydrogen atom (H) was documented as 1.007825 [57]; there were only 0.07–0.17% relative deviations between the experimental value and documental value. Undoubtedly, fragment *m/z* 299 could be considered a result of 2H-loss from fragment *m/z* 301. From the redox chemistry perspective, H-loss means oxidation. Therefore, fragment *m/z* 299 (i.e., quercetin *ortho*-quinone) can be regarded as the oxidized form of fragment *m/z* 301 (i.e., quercetin).

Fragment *m/z* 301 further released fragment *m/z* 151 via another Diels-Alder cleavage (Figure 7). In comparison, the fragment *m/z* 299 underwent decarbonylation twice, to generate *m/z* 272 and 243. The latter decarbonylation was assigned at the 4”-position rather than the 4”’-position (Figure 7); this was because standard quercetin was observed to produce a corresponding fragment—*m/z* 245 (Figure 6C). Subsequently, the *m/z* 243 experienced two α-cleavages (homolyzing) to lose the 1”-O atom and release *m/z* 227 (Figure 7). The experimental value of O-loss was 15.9956 (243.0289–227.0333, Figure 6F), while the documental relative mass of the O atom was 15.994915 [57].

In summary, MS elucidation revealed that quercetin interacted with DPPH^●^ to yield its dimer metabolite QDAD. Considering previous studies related to DPPH^●^ hydrogen-abstraction reactions [68,69] and quercetin dimerization reactions [23,70,71,72], the whole process can be proposed as shown in Figure 8.

It should be noted that, firstly, there was no so-called *syn* dimer because of the hinderance from head-head addition (Figure 8B). This phenomenon is termed “regioselectivity” [23,24,27,70,71]. Regioselectivity is supported by recent evidence showing that callus cultures could produce five *anti* Diels-Alder dimers (dysoverines A–E) and no *syn* Diels-Alder dimers in *Dysosma versipellis* [22].

Secondly, as shown in Figure 1C, quercetin is a planar molecule. If quercetin *ortho*-quinone concertedly (not stepwise) attacks quercetin below the plane in the head-tail direction, it gives a (*2R,3R*)-*anti*-dimer (Figure 8B bottom left and Figure 1D). If quercetin *ortho*-quinone concertedly attacks quercetin above the plane in the head-tail direction, it yields a (*2S,3S)*-*anti*-dimer (Figure 8B bottom right). Therefore, there is neither a *2S,3R*-configuration nor *2R,3S*-configuration, meaning that even head-tail addition has so-called stereoselectivities [73,74]. Nevertheless, the occurrence of a below-plane attack is of an equal likelihood to that of an above-plane attack, resulting in an equal chance of yielding a (2*R*3,*R*)-*anti*-dimer or (2*S*,3*S*)-*anti*-dimer. The two dimers exactly consist of a pair of racemates. Therefore, the correct and complete nomenclature of QDAD is “(±) quercetin Diels-Alder *anti*-dimer”, namely, (±) QDAD. As such, the previous nomenclatures are inappropriate, including “quercetin dimer” [27], “quercetin homodimer” [72], “quercetin heterodimer” [23,24], “C_2_-O-C_4′_*/C_3_-O-C_3′_* quercetin dimer” [70], and “1,3,11a-trihydroxy-9-(3,5,7-trihydroxy-4H-1-benzopyran-4-on-2-yl)-5a-(3,4-dihydroxyphenyl)-5,6,11-hexahydro-5,6,11-trioxanaphthacene-12-one” [71]. For convenience, the study preferred to simply refer to (±) quercetin Diels-Alder *anti*-dimer as “QDAD”.

Thirdly, it is possible, owing to this similarity, that the resolution of enantiomers of (±) QDAD is very difficult and the pair of racemates are used together. In fact, the pair of racemates share an identical CAS (Chemical Abstracts Service) number (167276-19-9). However, a similar pair of racemates has been found in *Dysosma versipellis* [22].

It is clear and interesting that quercetin can be converted into QDAD via a free radical oxidation (or enzymic oxidation) reaction and subsequent Diels-Alder addition reaction; the oxidation and Diels-Alder reactions can also characterize the cellular metabolism of quercetin. However, in the MS field, QDAD can be disconnected at the original Diels-Alder addition sites.

As shown in Figure 2, QDAD possessed a lower ferroptosis-inhibition level than its precursor quercetin. QDAD was also indicated to be a weaker antioxidant than its precursor by PTIO^●^-scavenging, Fe^3+^-reducing, Cu^2+^-reducing, DPPH^●^-scavenging, and linoleic acid emulsion (Table 1 and Appendix A) assays. The consistence between the data from the ferroptosis-inhibition assay and the antioxidant assay further supported that the ferroptosis-inhibition effects are highly related to their antioxidant actions.

However, there is seemingly a paradox. As shown in Figure 1, quercetin showed five phenolic -OHs (including one catechol moiety) and QDAD presented seven phenolic -OHs (including one catechol moiety) (Figure 1). The increase in phenolic -OHs is well-known to enhance the antioxidant effect. Hence, QDAD should be a stronger antioxidant than its precursor quercetin. Herein, we have tried to use redox chemistry to explain the paradox. From the perspective of redox chemistry, quercetin *ortho*-quinone is virtually an oxidized form of quercetin. The *ortho*-quinone moiety can be achieved through catechol oxidation by multiple pathways, including electron-transfer (ET) *plus* proton-transfer (PT) [75,76,77,78], hydrogen-abstraction [79,80,81], the ET pathway alone [82,83,84], or even enzymatic oxidation [25,26]. Therefore, it is very difficult for quercetin *ortho*-quinone to be further oxidized, suggesting that quercetin *ortho*-quinone is a weak antioxidant [79]. The metabolite QDAD can be regarded as a hybrid comprising a quercetin moiety and a quercetin *ortho-*quinone moiety. Definitively, the relative antioxidant activity of QDAD was lower than that of its precursor quercetin and higher than that of quercetin *ortho*-quinone.

As mentioned above, catecholic flavonol is an important family of natural products [33,34]. However, some catecholic flavonols were recently found to co-exist with the relevant Diels-Alder *anti*-dimers in the same plant, which was *Dysosma versipellis* [22]. As seen in Figure 9, these flavonols included kaempferide, podoverine A, kaempferol, and 3-*O*-methylquercetin; their relevant Diels-Alder dimers, however, were numbered as dimers **1–5** in the study. Dimer **1** can be regarded as a Diels-Alder addition product between kaempferide and 3-*O*-methylquercetin; herein, 3-*O*-methylquercetin offers the *ortho-*quinone moiety. Dimers **2–3** are the Diels-Alder addition products between kaempferide and podoverine A, and between kaempferol and podoverine A, respectively. Herein, podoverine A acts as the supplier of the *ortho-*quinone moiety. In addition, podoverine A also provides the *ortho-*quinone moiety for dimer **4.** Finally, dimer **5** is constructed using a kaempferol moiety. The co-existence and high relevance between catecholic flavonols and the Diels-Alder dimers have further indicated that the Diels-Alder dimers were the metabolites of catecholic flavonols in plants.

During plant metabolism, quite a few factors can oxidize catecholic moieties to *ortho*-quinone, including bio-enzymes (e.g., catechol oxidase and aurone synthase) [25,26,85], iron [83,86], LPO [86], and free radicals [75]. Of these, iron and LPO are relevant to ferroptosis. Undoubtedly, the formation of *ortho*-quinone with π-π conjugation has facilitated Diels-Alder dimerization reactions [22,23,24,27,70,71,72].

Therefore, the comparison of QDAD and quercetin in the study has provided a generalized rule to predict their relative ferroptosis-inhibition and antioxidant effects compared to those of their parent flavonols. In line with this rule, we can easily judge the relative ferroptosis-inhibitory (or antioxidant) effects of 3-*O*-methylquercetin vs. dimer 1, and 3,3′,4′-trihydroxyflavonol vs. its Diels-Alder dimer (6) (Figure 10) [72].

## 4. Conclusions

Catecholic flavonol quercetin can inhibit erastin-induced ferroptosis in bmMSCs, possibly through antioxidant pathways. During the antioxidant process, quercetin may convert into its Diels-Alder dimer metabolite QDAD. However, QDAD possesses weaker anti-ferroptosis and antioxidant activities than its precursor quercetin. The comparison of quercetin and QDAD has provided a generalized rule to predict the relative anti-ferroptosis and antioxidant levels of catecholic flavonols and their Diels-Alder dimer metabolites.

## Figures and Tables

**Figure 1 antioxidants-09-00205-f001:**
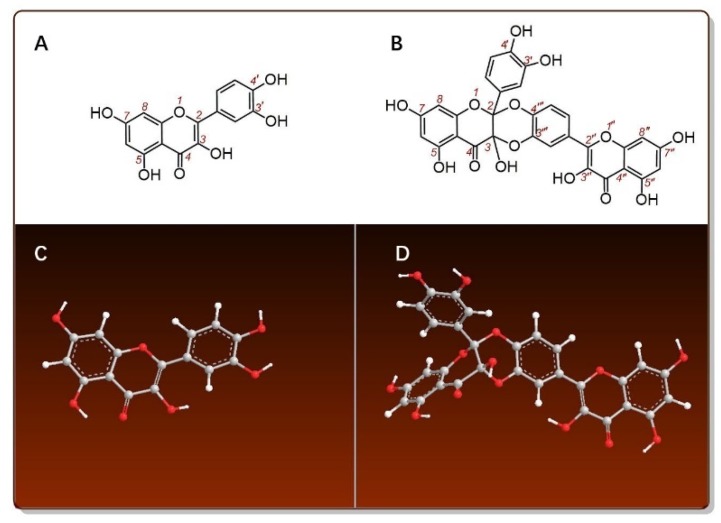
The structures and molecular models of quercetin and quercetin Diels-Alder *anti*-dimer (QDAD): (**A**) structure of quercetin; (**B**) structure of QDAD; (**C**) molecular model of quercetin; (**D**) molecular model of *2R,3R*-QDAD. The molecular model was created based on the preferential conformation by using Chem3D Pro 14.0 (PerkinElmer, Waltham, MA, USA).

**Figure 2 antioxidants-09-00205-f002:**
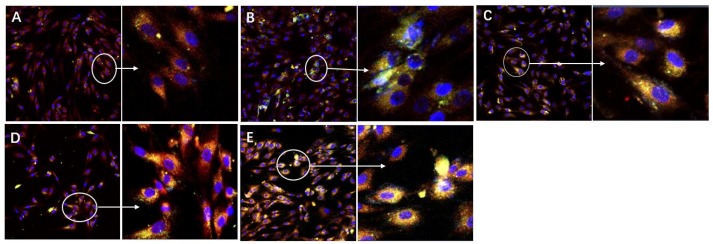
Fluorescence images of normal and ferroptotic bone marrow-derived mesenchymal stem cells (bmMSCs). (**A**) Control group (bmMSCs with no erastin treatment); (**B**) model group (erastin-treated bmMSCs); (**C**) positive control group (bmMSCs treated with erastin + ferrostatin-1); (**D**) quercetin sample group (bmMSCs treated with erastin + quercetin); (**E**) QDAD sample group (bmMSCs treated with erastin + QDAD). Lipid peroxidation (LPO) accumulation was probed using 4,4-difluoro-5-(4-phenyl-1,3-butadienyl)-4-bora-3a,4a-diaza-s-indacene-3-undecanoic acid (C11-BODIPY) and is depicted in green; cell nuclei were stained using DAPI (4‘,6-diamidino-2-phenylindole) and are shown in blue; and mitochondria were stained using BBcell Probe^TM^ and are shown as red fluorescence.

**Figure 3 antioxidants-09-00205-f003:**
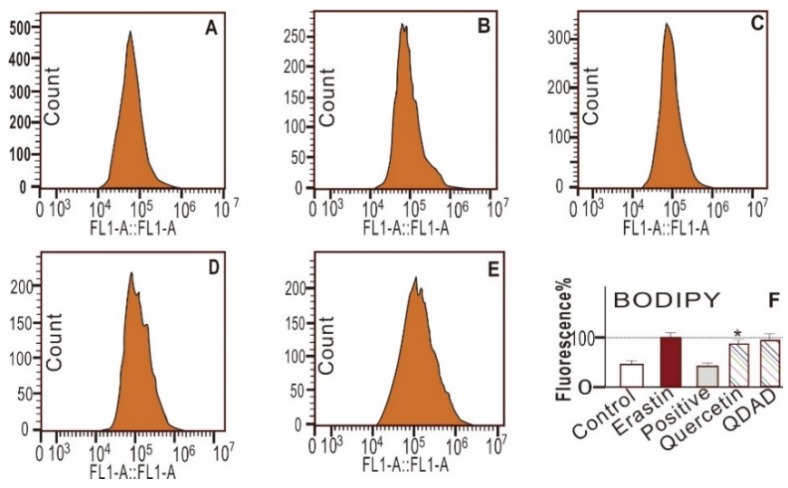
Representative flow cytometry analysis of the 4,4-difluoro-5-(4-phenyl-1,3-butadienyl)-4-bora-3a,4a-diaza-s-indacene-3-undecanoic acid (C11-BODIPY) fluorescence intensity (**A**–**F**) and the percentage of the relative mean fluorescence intensity of C11-BODIPY (**F**) of bmMSCs. (**A**) Control group; (**B**) model (erastin) group; (**C**) positive control (erastin + ferrostatin-1) group; (**D**) erastin + 0.03 μM quercetin; (**E**) erastin + 0.03 μM QDAD; (**F**) relative percentage of the mean fluorescence intensity of experimental groups. * The quercetin-treated samples (0.03 μM) have a significantly (*p* < 0.01) lower fluorescence than the erastin-treated group. Other groups do not show a significant decrease (*p* > 0.01). FL1-A (fluorochromes emitting A channel).

**Figure 4 antioxidants-09-00205-f004:**
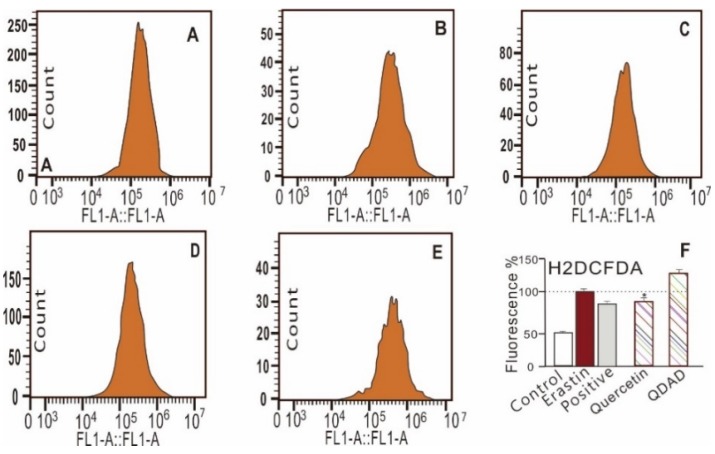
Representative flow cytometry analysis of the 2′,7′-dichlorodihydrofluoroscein diacetate (H2DCFDA) fluorescence intensity (**A**–**E**) and the percentage of the relative mean fluorescence intensity of H2DCFDA (**F**) of bmMSCs. (**A**) Control group; (**B**) model (erastin) group; (**C**) positive control (erastin + ferrostatin-1) group; (**D**) erastin + 0.03 μM quercetin; (**E**) erastin + 0.03 μM QDAD; (**F**) relative percentage of the mean fluorescence intensity of experimental groups. * The quercetin-treated sample group (0.03 μM) has a significantly (*p* < 0.01) lower fluorescence intensity than the erastin-treated samples. H2DCFDA: 2′,7′-dichlorodihydrofluoroscein diacetate; FL1-A: fluorochromes emitting A channel.

**Figure 5 antioxidants-09-00205-f005:**
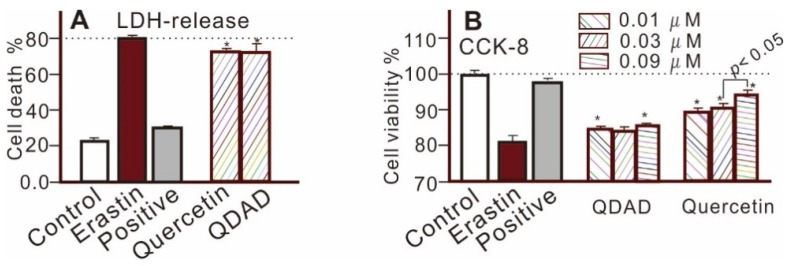
The lactate dehydrogenase (LDH) release assay (**A**) and cell counting kit-8 (CCK-8) assay (**B**) of erastin-induced ferroptotic bmMSCs. The erastin group is the model group, while erastin + ferrostatin-1 is the positive control group. * *p* < 0.05, compared with the erastin-treated group. Significant (*p* < 0.05) difference is shown between the 0.03 μM quercetin-treated group and the 0.09 μM quercetin-treated group in **B**.

**Figure 6 antioxidants-09-00205-f006:**
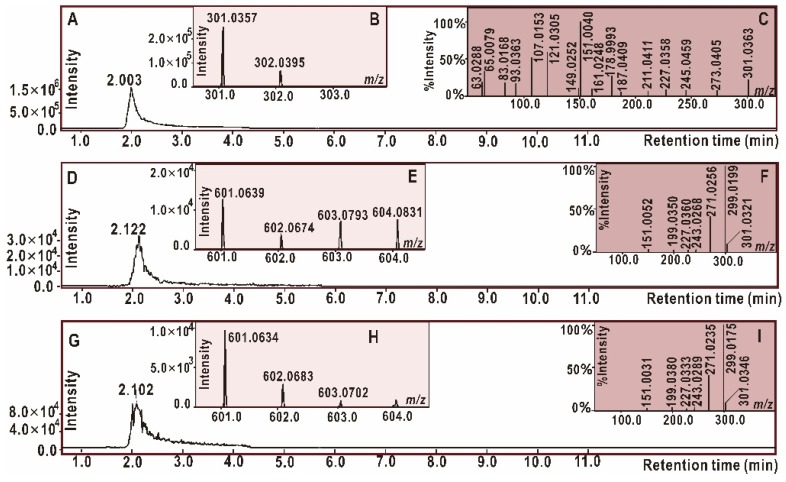
Main results of ultra-performance liquid-chromatography coupled with electrospray-ionization quadrupole time-of-flight tandem mass spectrometry (UHPLC-ESI-Q-TOF-MS/MS) analysis. (**A**) Chromatogram of quercetin when the formula [C_15_H_10_O_7_−H]^−^ was extracted. (**B**) Primary MS spectra of quercetin. (**C**) Secondary MS spectra of quercetin. (**D**) Chromatogram of possible dimeric products of quercetin-quercetin when the formula [C_30_H_18_O_14_−H]^−^ was extracted. (**E**) Primary MS spectra of possible dimeric products. (**F**) Secondary MS spectra of the radical adduct formation (RAF) product. (**G**) Chromatogram of standard QDAD when the formula [C_30_H_18_O_14_−H]^−^ was extracted. (**H**) Primary MS spectra of standard QDAD. (**I**) Secondary MS spectra of standard QDAD.

**Figure 7 antioxidants-09-00205-f007:**
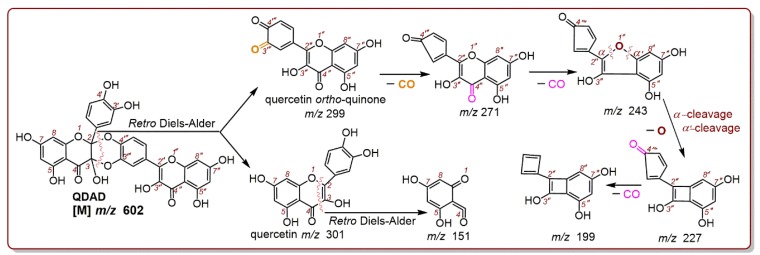
The proposed MS elucidation of QDAD (the accurate *m/z* values are simply expressed as integers; electron transfer has not been marked. Other reasonable cleavages should not be excluded in the MS elucidation).

**Figure 8 antioxidants-09-00205-f008:**
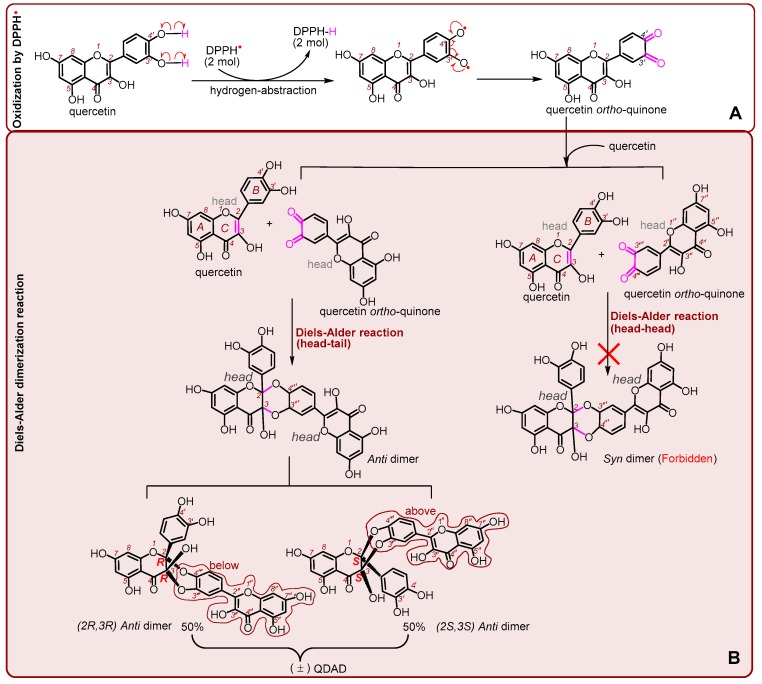
Scheme for the cause of nomenclature of the “(±) quercetin Diels-Alder *anti*-dimer”. (**A**) quercetin oxidation by the DPPH^●^ radical; (**B**) Diels-Alder dimerization reaction (the single-barbed curved arrow indicates one electron transfer in Figure A).

**Figure 9 antioxidants-09-00205-f009:**
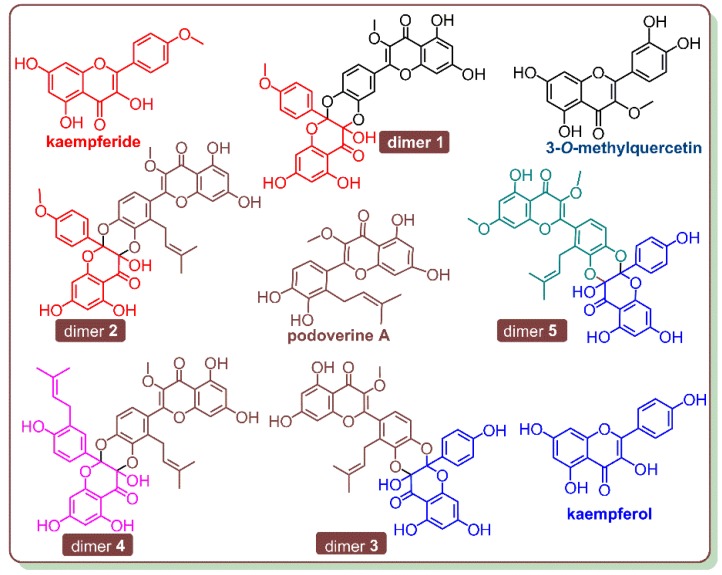
The structures of catecholic flavonols and the relevant Diels-Alder dimers in *Dysosma versipellis.*

**Figure 10 antioxidants-09-00205-f010:**
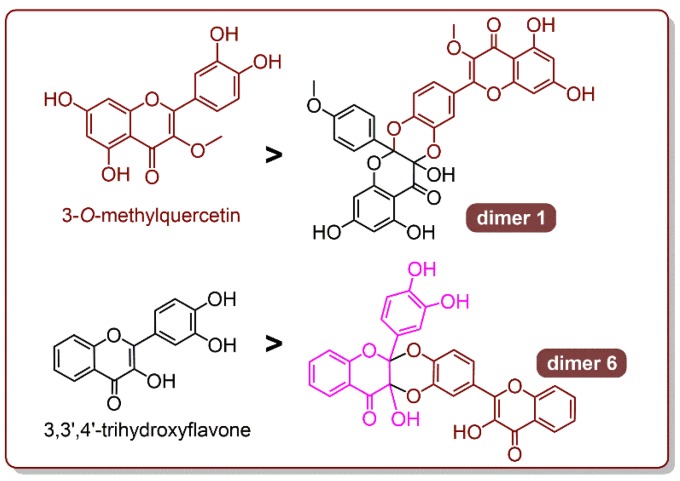
The prediction of relative antioxidant levels for catecholic flavonols and the Diels-Alder dimers.

**Table 1 antioxidants-09-00205-t001:** The IC_50_ values of QDAD and quercetin in various antioxidant measurements (μM).

Antioxidant Measurements	QDAD	Quercetin	Trolox	Ascorbic Acid
PTIO^●^-scavenging	181.2 ± 10.9 ^b^	147.3 ± 2.0 ^a^	646.0 ± 98.2 ^d^	457.4 ± 24.2 ^c^
Fe^3+^-reducing	40.1 ± 1.2 ^b^	15.2 ± 1.0 ^a^	43.8 ± 5.6 ^c^	91.7 ± 4.4 ^d^
Cu^2+^-reducing	43.3 ± 1.2 ^b^	37.4 ± 3.0 ^a^	115.2 ± 11.6 ^c^	183.5 ± 8.2 ^d^
DPPH^●^-scavenging	20.3 ± 2.0 ^b^	10.9 ± 1.1 ^a^	35.6 ± 4.5 ^c^	51.7 ± 0.7 ^d^
Lipid peroxidation-scavenging	15.5 ± 1.2 ^c^	2.2 ± 0.02 ^a^	136.0 ± 12.3 ^d^	3.0 ± 0.1 ^b,^*

The IC_50_ value is defined as the lowest concentration with 50% radical inhibition or relative reducing power, calculated by linear regression analysis, and expressed as the mean ± SD (*n* = 3). The linear regression was analyzed using Origin 6.0 professional software. The IC_50_ values with different superscripts (^a^, ^b^, ^c^ and ^d^) in the same row are significantly different (*p* < 0.05). Trolox and ascorbic acid were used as the positive controls. All dose-dependent curves are given in Appendix A. QDAD, (±) quercetin Diels-Alder *anti*-dimer. *, Butylated hydroxyanisole (BHA) instead of ascorbic acid.

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
