# Peer review of "Inhibitory Effect and Mechanism of Action of Quercetin and Quercetin Diels-Alder anti-Dimer on Erastin-Induced Ferroptosis in Bone Marrow-Derived Mesenchymal Stem Cells"

_antioxidants, 2020, doi:10.3390/antiox9030205_

Round 1
Reviewer 1 Report
The authors have not addressed my questions. The protective effect of both, Quercetin and QDAD, is weaker than Ferrostatin-1. Although Quercetin showed antioxidant activity, its effect over cell death is not consistent.
Reviewer 2 Report
Dear Editor and Authors
The manuscript has been improved and I recomend its acceptance.
Sincerely
Author Response
Thank you very much for the fair assessment!
Reviewer 3 Report
After reading and reading, I still do not get the idea of only comparing to QDAD.Needs this needs further explanation.
Further, comaparison with related flavonol structures, whose antioxidant capacity is well described the literature would have been (more ?) helpful for identifying structural features.
I am not convinced with following aspects. Those need either more explanation or referencing to literature:
Introduction, line 89: "DPPH• radical, a mimic of ROS or LOO• radical;" Too simplified....
Lines 275-277: "and the corresponding Diels-Alder anti-dimer metabolites may distribute as widely as these catecholic flavonols. One evidence of this fact is that so many flavonol Diels-Alder anti-dimers can be found in the same plant" Here a discussion on concentrations would have been helpful.
Aim is not clearly presented. Usually, at the end of the introduction, aims are clearly mentioned and/or hypotheses claimed. Here, it is more like a summary with kind of conclusion.
Literature format is quite mixed. Many orthographical mistakes...sometimes too many blanks...different styles in citing....in parts , it looks quite weird...
Round 2
Reviewer 3 Report
Manuscript has been significantly improved during the revision. However, some minor mistakes in language and style are still there, but maybe these will be treated by the editorial office or publisher...
This manuscript is a resubmission of an earlier submission. The following is a list of the peer review reports and author responses from that submission.
Round 1
Reviewer 1 Report
Li et al demonstrate that Quercetin prevented Erastin-induced ferroptosis in Bone Marrow-derived mesenchymal cells. However, the experiment that demonstrates this is not convincing, and more experiments are necessary to confirm this. Actually, the paper is focused in the chemical approached to explore the effect of quercetin.
The authors should quantify the Bodipy staining (e.g. by flow cytometry) to confirm the anti-ferroptotic effect of quercetin. Moreover, cell death should be quantified by a more specific technique as measurement of cytotoxicity by LDH release. The protective effect of quercetin should be compared with a specific inhibitor of ferroptosis as Ferrostatin-1. The antioxidant effect of quercetin over other sources of oxidative stress has to be measured in cultured cells, as total ROS with H2DCFDA probe or mitochondrial ROS. In figure 2 the authors do not indicate the number of independent experiments neither the statistic for this result. The scavenging effect of quercetin has to be measured with specific substrates of ferroptosis as peroxyl radicals.
Reviewer 2 Report
English language
In PAGE 2 – lines 3 and 4
would be adequate to use: ON THE ONE HAND positive modulation (…). ON THE OTHER HAND negative modulation (…)
In PAGE 4 –2nd paragraph, line 4
Would be better to use: neurone, neuronal cell or nerve cell instead of neuro-cell
In PAGE 7 –2nd paragraph, line 2
enzymatic oxidation instead of enzymic oxidation
Overall, would be better to use oxidation instead of oxidization
In PAGE 2 –2nd paragraph, last sentence: provide some examples of phenolic antioxidants
It would be nice to add a table that summarize different studies (performed in different cell lines) on the mechanism of action of quercetin on ferroptosis, as it would give a global vision of the compound.
Reference list:
Number 39 and 41 are the same / Number 40 and 42 are the same
According to authors quercetin and its dimer metabolite usually co-exist in the same plant. Taking this into account, it would be interesting for a future research to compare the quercetin to another phenolic compound from catecholic flavonol family.
ABOUT THE MANUSCRIPT OVERALL:
The research of Li et al. uses an interesting cell model (bone marrow-derived mesenchymal stem cell) with impact on several knowledge areas such as cell biology, chemical biology, metabolomics, stem cell transplantation and antioxidant chemistry.
The study also provides interesting data not only to characterize the cellular metabolism of quercetin and prove that quercetin can be converted into QDAD via free radical oxidation, but also to elucidate that ferroptosis-inhibition effects of quercetin are highly related to their antioxidant actions.
The models and methods used are accurate.
Reviewer 3 Report
Dear Authors and Editor,
In the reviewed manuscript titled Inhibitory Effect and Mechanism of Action of Quercetin on Erastin-induced Ferroptosis in Bone Marrow-derived Mesenchymal Stem Cells”, it was stated that the title of the article does not have relevant content. The effect of quercetin and QDAD on inhibition of bone marrow key feroptosis has not been proven. According to the opinion of the Reviewer only a divagation on the antioxidant properties of quercetin and QDAD and mechanisms of their antioxidant properties were shown not any biological activity. In the Reviewer opinion the abstract is not presenting the real content of the manuscript. That is why the Reviewer recommends rejection of the manuscript.
Moreover,
Figure 2a- poor visibility.
Figure 3 – explain why the retention time quercetin as the same as QDAD
Yours sincerely